# A Modeling Platform for Landslide Stability: A Hydrological Approach

**Mohsen Emadi-Tafti** [1,*] and **Behzad Ataie-Ashtiani** [1,2]

1   Department of Civil Engineering, Sharif University of Technology, Tehran 11365-11155, Iran; ataie@sharif.edu
2   National Centre for Groundwater Research and Training, College of Science and Engineering, Flinders University, Adelaide, SA 5001, Australia
*   Correspondence: moh.emadi@gmail.com

**Abstract:** Landslide events are among natural hazards with many fatalities and financial losses. Studies demonstrate that natural factors such as rainfall and human activities such as deforestation are important causes of triggering a landslide. In this study, an integrated two-dimensional slope stability model, SSHV-2D, is developed that considers various aspects of hydrological effects and vegetation impacts on the stability of slopes. The rainfall infiltration and water uptake of roots change the water content of the unsaturated zone. The temporal and spatial distribution of water content is estimated in the hydrological unit of the developed model. The vegetation unit of the model considers interception loss due to the existence of canopies and trunks, soil reinforcement effect by roots, root water uptake, the impact of root on hydraulic conductivity, and the influence of vegetation weight on slope stability. Benchmark problems with and without vegetation are solved for the model verification. The analyses demonstrate that the consideration of matric suction in the unsaturated zone can increase the safety factor more than 90%. It is also observed that the existence of trees with high density on a slope can increase the factor of safety about 50% and prevent shallow landslides. The present model is a platform for further development of more comprehensive and elaborative slope stability models.

**Keywords:** water flow in unsaturated zone; Richard's equation; slope stability; vegetation; hydrology

## 1. Introduction

Identification and investigation of landslide events due to the lack of slope stability are important due to possible economic losses and fatalities. Although landslides are less common compared with other natural hazards such as earthquakes and storms [1,2], available databases confirm the significance of losses due to landslide events. Froude and Petley [3] reported that an average of 374 fatal non-seismic landslides with more than 4300 fatalities occurs annually worldwide. Aleotti and Chowdhury [4] mentioned that among all the natural phenomena, the landslide has the highest fatalities after drought, hurricane, earthquake, and flood. Moreover, landslides can damage infrastructures such as roads, railways, pipelines, buildings, etc. [5,6]. In addition to direct damages, landslide may create indirect incidents such as floods which are caused by impulsive waves in dam reservoirs or lakes [7]. The slide in the Vajont reservoir in Italy associated with 2500 fatalities is a sample of a landslide-triggered wave [8]. Landslides account for global losses of around 20 billion USD annually [5,6]. Nearly 5% of the total world population lives in the regions with landslide potential [9]. These statistics demonstrate the importance of investigating and modeling landslides.

Landslides occur due to natural causes, human activities, or a combination of them [5]. Rainfall increases the pore water pressure which can increase landslide probability by 90% [5,10,11]. Climate change has resulted in rainfalls with less frequency but higher intensity [12]. This can worsen the

landslide problem. Human activities and deforestation also play a role in landsides [5]. Economic developments and activities such as construction, agriculture, logging, mining, etc., have changed the land cover [13]. Meehan [14] stated that the clear-cutting of forests increases the frequency of mass movement more than six times. The massive landslides that occurred in November 2004 in the Philippines are a sample of landslides which are triggered by heavy rainfall after spread deforestation [15]. These findings demonstrate that considering hydrological and vegetation effects on slope stability is important.

Slope stability control with vegetation was performed as early as the 16th century in China for dam stabilization [16]. However, the incorporation of vegetation effects on slope stability analysis was introduced in the 1960s [17]. The most common effect of vegetation was the mechanical reinforcement of soil due to the existence of roots which were considered as an additional cohesion to soil shear strength. The suggested model by Waldron [18] and Wu et al. [19] is the first and most used model for this effect. In addition to the reinforcement effect of the roots, water uptake of roots also influences slope stability. Gardner [20] was the first researcher who presented a mathematical model for water uptake. Afterward, Feddes et al. [21], Prasad [22], Li et al. [23], and etc. developed further models. Another effect of vegetation is the interception of rainfall. Canopy and trunks of trees store a portion of precipitation that eventually evaporates. This term is named interception loss and was investigated by Rutter et al. [24,25] and Gash [26] for the first time. The sparse Rutter model [27] and the sparse Gash model [28] are the most commonly used models.

Analysis of slope stability started in the 1920s. Different schemes are available for slope stability analysis and the limit equilibrium methods [29–32] are commonly used. In slope stability analysis the effective normal stress on slip surface is a key parameter. Typically, the effective normal stress in the unsaturated zone (above the water table) is considered equal to total stress. Bishop [33] introduced the matric suction concept in the unsaturated zone and stated that effective normal stress in this zone is greater than the total normal stress. This concept increased the soil strength and the factor of safety in stability analysis of slopes. Matric suction is a function of soil moisture. Thus, infiltration of rainfall and change in water content affects the stability of slopes. Darcy [34] proposed a rule for water flow in the saturated soil. Then Richards [35] developed a partial differential equation for water movement in the unsaturated zone. The solution of Richards' equation is soil moisture distribution in the domain which can be used in slope stability analysis.

There are a few existing models that consider hydrological and/or vegetation impacts on slope stability, including the following software: CHASM [36,37], SLIP4EX [38], modified RSLOPE [39], HYDROlisthisis [40], and SHIA-Landslide [41]. Also, the GeoStudio software package (Geostudio 2012) can model features of hydrological and vegetation effects by linking results of SEEP/W and VADOSE/W with SLOPE/W. However, some of these models do not consider both hydrology and vegetation aspects, and all of them have limitations. Among the mentioned models, CHASM and the GeoStudio software package are more comprehensive. CHASM assumes that rainfall infiltrates in the vertical direction only and cannot model water movement in two dimensions. On the effects of vegetation, CHASM ignores water uptake and considers simple root reinforcement only. Moreover, the interception of the rainfall by the canopy is disregarded in all models. Another important limitation is that some of these programs are commercial products and there is no access to their source for adding new features.

In recent years, there have been significant advances in different fronts such as efficient and robust modeling of unsaturated flow, modeling of vegetation influences on slope stability, root uptake, and root reinforcement which can improve slope stability analysis. The objective of this work is to apply state-of-the-art knowledge and methods in this regard for developing a numerical model for the slope stability analysis with a hydrological and vegetation effects approach. The model will be a platform for further developments in slope stability analysis. This model is able to simulate rainfall, water movement in the soil (e.g., fluctuation of the water table, increase or decrease of pore water pressure), and consider different impacts of vegetation cover (e.g., root reinforcement, root water uptake, and interception loss) on slope stability. The final output of the program is the safety factor of the slope,

which can be used to determine whether the slope is critical or not. The model can be used to predict the impacts of other changes such as deforestation and climate change on the stability of existing slopes.

This paper is organized as follows. First, the structure of the model is presented which includes three main features of slope stability, hydrology, and vegetation impacts. In the second section, mathematical formulations governing each module are presented. In the following section, it has explained how to discretize equations and what are numerical schemes. Finally, the benchmark problems are solved by the program for verification and the influences of considering hydrological and vegetation impacts on slope stability are reported.

## 2. Model Structure

The presented program is a two-dimensional slope stability model with hydrological and vegetation effects with the name SSHV-2D. This program was developed using Fortran95 which works in the Microsoft Windows environment.

The plants' influence on slope stability has many different features. The most common impact of vegetation is mechanical reinforcement of soil by roots which enhances soil shear strength. On the other hand, due to transpiration, the plant roots absorb the water content of soil and increase matric suction which leads to improved stability of a slope. In return, the infiltration of rainfall increases the water content of the soil, but the holding capacity of the vegetation system can reduce the rate of rainfall reaching the ground and its negative effects. Moreover, the weight of trees and the wind influences the equilibrium of the slope which is taken into account in the model. Figure 1 shows a schematic of the conceptualization of the model and the processes influencing the slope stability assessment.

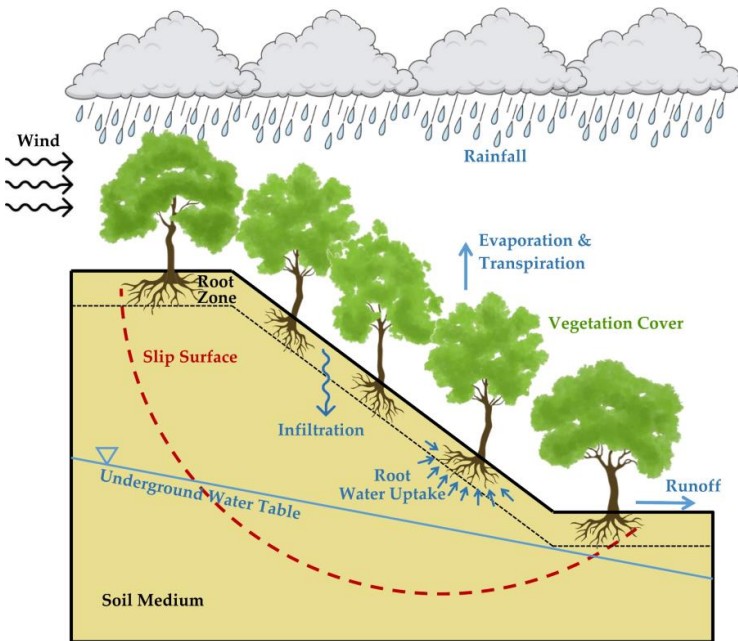

**Figure 1.** Schematic of vegetated slope stability analysis.

The mentioned physical processes are modeled in a four-stage procedure framework which is presented in Figure 2. In the following subsections, the operation of each unit is described.

### 2.1. Primary Preparation Unit

In the first stage, the input parameters including soil properties, hydrological parameters, vegetation data, the geometry of slope, and required information for numerical modelings such as mesh size, initial condition, and boundary condition are collected. In this version, the input parameters are taken as text files, which can be performed through the user interface in subsequent versions.

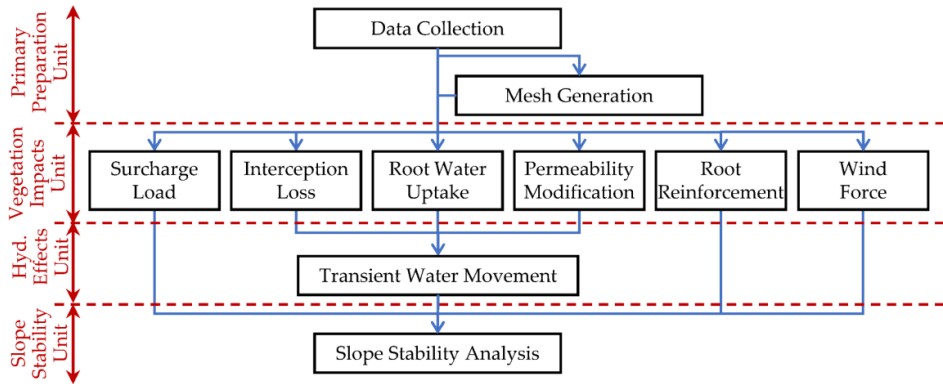

**Figure 2.** Flowchart of the developed model.

The spatial discretization of the domain is done in this stage using rectangular cells for the finite difference method (FDM). The user determines mesh dimensions. After the domain converts to cells, the "properties matrix" is created. The matrix includes the properties of soil for each cell such as specific weight, cohesion, friction angle, and hydraulic conductivity. In multi-layer domains, depending on cell positions in the soil medium, the properties matrix is complemented by the properties of the related soil layer.

The heterogeneity of soil is an important issue that has a not evanescent impact on the stability of slopes [42,43]. In this stage of model development, the considered problems are limited to the cases with homogeneous soils. Heterogeneous cases are considered in future works.

### 2.2. Vegetation Impacts Unit

There are six sub-modules for the vegetation effects unit, of which three of these (interception loss, root water uptake, and permeability modification) influence on the hydrology unit and three others (root reinforcement, surcharge load, and wind force) affect the stability unit directly. During a rainfall event, a portion of it is intercepted as it stays on the canopy and trunks and it is evaporated. The sum of throughfall (water falling from canopy) and stemflow (water flowing down the tree trunks) is the effective rainfall. The effective rainfall infiltrates to soil and the surplus goes into the runoff. Another vegetation effect is water uptake done by the roots. The water uptake acts as a sink term in the cells where the root is. The third hydrological impact of vegetation roots is to increase soil permeability and the velocity of water infiltration. This impact can be accomplished by modifying the hydraulic conductivity element of the root zone cells in the properties matrix.

The other three features are the mechanical impacts of vegetation. The vegetation roots improve the mechanical strength of the soil. The tensile strength of the root enhances soil shear strength and leads to more stability. The improvement is implemented as an additional cohesion to the soil. The vegetation weight is the fifth impact of vegetation that alters the shear and normal stress on the slip surface. The surcharge due to vegetation has two opposite effects, both increasing the driving and the resistance forces. The wind, depending on the flow direction, can play both a beneficial and an adverse role. Since wind flow direction can be changed, considering downslope direction, which has a negative effect on stability, is conservative.

### 2.3. Hydrological Effects Unit

The canopy intercepts a part of gross precipitation that is considered in the vegetation unit. The groundwater hydrology calculates the movement of water and moisture distribution in the soil medium. Water infiltrates in soil and causes a change in soil moisture. The increasing soil water content decreases the soil matric suction which reduces the shear strength of the soil and, finally, decreases the safety factor of the slip surface. Again, the root water uptake (one of the vegetation effects) is a positive parameter in stability of the slope, because it removes moisture and increases the matric suction.

*2.4. Slope Stability Analysis Unit*

Slope stability analysis is performed by the limit equilibrium approach. The circular and non-circular slip surfaces can be considered in this unit. The unsaturated soil analysis and determination of matric suction come from the hydrology unit, and the reinforcement of soil and the surcharge load from the vegetation unit determine the effect on the slope stability unit.

## 3. Mathematical Formulation

*3.1. Vegetation Module*

The influences of vegetation on slope stability can be categorized into two groups: the root effects and the canopy effects. Greenway [17] mentioned these effects as shown in Table 1. In the following, these effects are described and related formulations are presented.

**Table 1.** Influences of vegetation on slope stability [17].

| Mechanism | Result | Influence |
|:---:|:---:|:---:|
| Root (underground) portion | | |
| Reinforcement and anchorage by root | Increasing soil shear strength | + |
| Root extracts moisture from the soil | Lower pore water pressure | + |
| Increasing hydraulic conductivity | Increased infiltration capacity | − |
| Canopy (aboveground) portion | | |
| Canopy intercepts and evaporates rainfall | Reduce rainfall for infiltration | + |
| Weight of trees surcharges the slope | Increasing normal stress | +/− |
| Vegetation exposed wind forces into the slope | Increasing driving force | − |

3.1.1. Reinforcement of Soil by Roots

The reinforcement of soil by the roots takes into account an increased cohesion to the cohesion of the soil.

$$c' = c'_s + c'_r \tag{1}$$

Here $c'_s$ ($ML^{-1}T^{-2}$) is the soil cohesion, $c'_r$ ($ML^{-1}T^{-2}$) is root contribution cohesion, and $c'$ ($ML^{-1}T^{-2}$) is the total cohesion in the root zone. The apparent cohesion due to existence of the roots may be considered as two main approaches: measurement approach and estimation approach. In the first, the increased cohesion is measured by field or laboratory shear tests (the difference between shear resistance of the root-permeated soil and bare soil shows root contribution cohesion). If this approach is considered, the SSHV-2D program gets a constant value as apparent cohesion caused by roots for each root-permeated soil layer.

In the estimation approach, the increased cohesion by roots is calculated by a relationship that includes root characteristics and likely soil properties. It is notable that many parameters of roots (such as the number of roots, diameter, length, direction of roots, tortuosity, root density, root architecture and morphology, branching characteristics, special distribution, tensile strength, Young's modulus, etc.) and many properties of soil (such as soil type, friction coefficient, water content, confining pressure, heterogeneity, etc.) are involved in increased cohesion by the roots, but all of them cannot be considered. The different models have been suggested for the quantification of the reinforcement effect of roots on the soil strength. The simple perpendicular root model is the first model developed by Waldron [18] and Wu et al. [19] The more important assumptions of the Wu/Waldron model (WWM) were that the failure mechanism of the roots is rupture (the pull-out failure mechanism was ignored) and all of the roots in the shear plane were considered at full tensile strength simultaneously [44]. In reality, the pull-out failure may happen if the retaining force is not enough. Also, the tensile stress in the roots at an instance in time is different and the roots break progressively. Therefore, the assumptions lead to overestimation of root contribution cohesion [45]. Two other fundamental models were introduced after WWM. These models, which considered a bundle of root instead of single roots, attempted

to account for progressive root breaking by gradually increasing of the load, are the fiber bundle model (FBM) [45] and the root bundle model (RBM) [46,47]. The FBM applies the load gradually and distributes it between the bundle of fibers (roots) evenly [45]. When the load in some fibers exceeds their maximum strength, the fibers break and their load redistributes between intact fibers. This process continues until all of the fibers break. But, the RBM is a strain-controlled loading model and considers the stretching and tortuosity of roots. Also, the pull-out resistance of roots and interaction of soil and root have been considered in the RBM [46,47].

In the SSHV-2D program, the modified WWM is used as the estimation approach. The WWM model has been commonly employed by researchers and the number of its input parameters is lower than other models, but many researchers have stated the model overestimates increased cohesion by the roots [45,48–52]. In other words, the estimated value by WWM is the maximum increased cohesion which in most cases cannot be achieved. Therefore, considering the reduction factor, the WWM is modified [53,54] and can be used in the analysis as below:

$$c'_r = k'' \left( 1.2 \sigma_r \frac{A_r}{A} \right) \tag{2}$$

where $\sigma_r$ (ML$^{-1}$T$^{-2}$) is the average of tensile stress in the root, $A_r$ (L$^2$) is the total cross-sectional area of roots, $A$ (L$^2$) is the cross-sectional area of soil in considered section, $A_r/A$ is so-called the root area ratio (RAR) and $k''$ is the reduction factor. A wide range of reduction factors reported by researchers is listed in Table 2. Wu [51] suggested using a reduction factor of 0.3–0.5 according to practical approaches.

The WWM considers the soil and the root as a homogeneous medium in the root zone, and is suitable for the little roots with small spacing [51]. In reality, the RAR decreases with depth, but in the current version of SSHV-2D, it is not possible to consider RAR changes in one layer. For modeling RAR variations, the root zone can be divided into several layers and the average RAR per layer is used to evaluate the additional cohesion.

Another model, which is available in the SSHV-2D program, is an individual root model. In this model, each root is modeled separately as a bar that is able to withstand shear and axial forces and interacts with soil frictionally. As it is shown in Figure 3a, in addition to the shear strength of root (Q) where the root intersects with slip surface, the end portion of the root which has crossed the slip surface ($L_{er}$) increases the soil resistance by providing an effective tension force [60]. The tension in the root (shown with F in Figure 3a) increases resistance through two mechanisms. If the effective length of the root ($L_{er}$) is less than a threshold length ($L_t$), then the root pullout resistance is dominant. Otherwise, the tensile capacity of the root determines additional cohesion by the roots [60]. The threshold length can be determined by the division of tensile capacity by the pullout resistance (see Figure 3b). The root direction is not necessarily perpendicular to the slip circle; the tension force will have a tangent and a normal component. The root contribution to the cohesion due to reinforcement can be expressed as [60]:

$$c'_r = \frac{\sum F \cos \theta + F \sin \theta \tan \phi + Q}{L_{arc\ slip}} \tag{3}$$

where $\theta$ [-] is the angle between root and tangent of slip circle and $L_{arc\ slip}$ [L] is the length of the slip arc. The summation sign in the above equation shows that the additional shear force must be considered for all trees. This method can be used if the location, length, inclination, tensile capacity, pullout resistance, and shear capacity of individual roots are known. In SSHV-2D it is assumed that the trees are at equal distances. So, having the distance between trees is sufficient to specify the location of trees. Root density (RD) is defined as the number of roots (or trees) per unit length of the slope. Accordingly, the distance between trees is equal to the reverse of the RD and the distance of the first tree is half of this value from the start point of the slope.

**Table 2.** The reduction factor reported by some researchers to the modified Wu/Waldron model (WWM).

| Reference | $k''$ | Method | Soil | Vegetation |
|---|---|---|---|---|
| Wu and Watson [52] | 0.33 | In-situ shear test | Silty sand | *Pinus radiata* |
| Operstein and Frydman [50] | 0.21 | In-situ and laboratory tests | Chalky and clay | *Alfalfa, Rosemary, Pistacia lentiscus, Cistus* |
| Pollen, et al. [44] | 0.34 | Direct shear-box test | Clayey-silt | Riparian vegetation (12 species) |
| Pollen and Simon [45] | Trees: 0.6–0.82 Grass: 0.48 | In comparison with FBM | Silt | trees: *Cottonwood, Sycamore, River birch, Pine, Black willow* grass: *Switchgrass* |
| Docker and Hubble [55] | 0.61–0.64 | In-situ shear test | Alluvial (loam and sandy loam) | *Casuarina glauca, Eucalyptus amplifolia, Eucalyptus elata,* and *Acacia floribunda* |
| Fan and Su [56] | Peak: 0.325 Residual: 0.35 | In-situ shear test | Sands mixed with silts | *Prickly sesban* |
| Bischetti, et al. [53] | 0.27–0.83 | Direct shear test | Various (gravel-sand mixture, clayey, silt, . . . ) | *European beech, Norway spruce, European larch, Sweet chestnut, European hop-hornbeam* |
| Mickovski, et al. [49] | ~0.75 | Direct shear test (laboratory) | Agricultural soil (71% sand, 19% silt, 10% clay) | Willow |
| Mao, et al. [57] | 0.55–1.0 | In comparison with FBM | Various (silt, silty-clay, coarse elements) | Norway spruce, Silver fir, European beech |
| Adhikari, et al. [58] | 0.35–0.56 | In comparison with FBM | Fine sand texture | *A. lentiformis, A. occidentialis, L. andersonii, L. tridentata* |
| Meijer, et al. [59] | 0.08, 0.225 | Corkscrew test (field and lab.) | Slightly clayey sand, sandy silt | *Blackcurrant* (shrub), *Sitka spruce* (tree) |

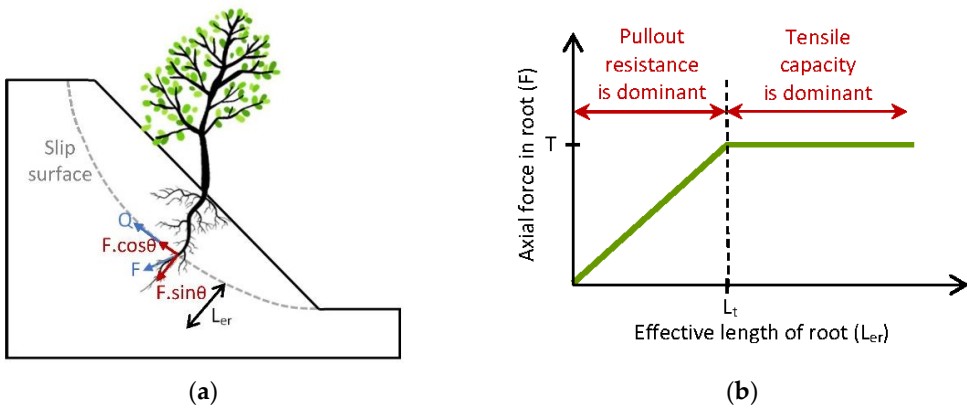

**Figure 3.** (**a**) Schematic view of soil–root interaction model; (**b**) the dominant mechanism in root reinforcement (adapted from Zhu, et al. [60]).

### 3.1.2. Increasing Permeability

The root of plants can increase the permeability of the soil in the root zone. In SSHV-2D, the user can introduce this phenomenon by entering a coefficient that is greater than one. Collison and Anderson [61] used the root area ratio (RAR) as the increasing coefficient. If the user does not define this coefficient, the default value of 1 is used.

### 3.1.3. Root Water Uptake

Plants extract water for photosynthesis and transpiration. The volume of water for photosynthesis is negligible compared with transpiration water [62], therefore, most of the water uptake is used for transpiration. The root water uptake can be computed from microscopic and macroscopic approaches. In the microscopic approach, each root branch is considered separately, but the measurement is difficult and not practical. In the macroscopic approach, the root system is considered as a single unit which is easier to implement [63].

The various proposed models for root water uptake are categorized into two groups: one-dimensional methods (1-D) and multi-dimensional methods. One-dimensional models only consider depth direction (e.g., [21,22]). It is appropriate for the analysis of lands uniformly covered with crops and cannot consider trees [64]. In the multi-dimensional methods, both lateral and vertical directions of root distribution are considered based on various shapes [65]. Fatahi [66] included semi-spherical, inverted cone, and cylindrical root shapes. In SSHV-2D code, it is assumed that the root follows an exponential distribution, having the maximum root density beneath the trunk, with an inverted cone shape, and constant root density on boundaries.

The most important parameters for root water uptake are transpiration rate, root length density, and matric suction in root location which is considered here. Other parameters such as permeability of soil and root, salinity stress, and frozen soil conditions are not considered. The root water uptake can be expressed as [66,67]:

$$S = \alpha(h) \cdot G(\beta) \cdot F(T_P) \tag{4}$$

where $\alpha(h)$ is the soil suction factor, $G(\beta)$ is the root density factor, and $F(T_P)$ is the potential transpiration factor. In this equation, $h$ is negative pore-water pressure, $\beta$ is the root length density, and $T_P$ is the potential transpiration rate.

The soil suction factor (that is called the reduction factor in some references) indicates that the maximum water uptake only occurs in a specified range. The different functions are proposed for the

soil suction factor (e.g., [21,68,69]). The most common formula was proposed by Feddes [70] which is expressed by:

$$\alpha(h) = \begin{cases} 0 & |h| \le h_1 \\ \frac{|h|-h_1}{h_2-h_1} & h_1 < |h| \le h_2 \\ 1 & h_2 < |h| \le h_3 \\ \frac{h_4-|h|}{h_4-h_3} & h_3 < |h| \le h_4 \\ 0 & h_4 < |h| \end{cases} \tag{5}$$

where $h_1$ (L), $h_2$ (L), $h_3$ (L), $h_4$ (L) are constant values. It is assumed that these points are known based on plant type and entered as input data into the program. This factor is shown in Figure 4.

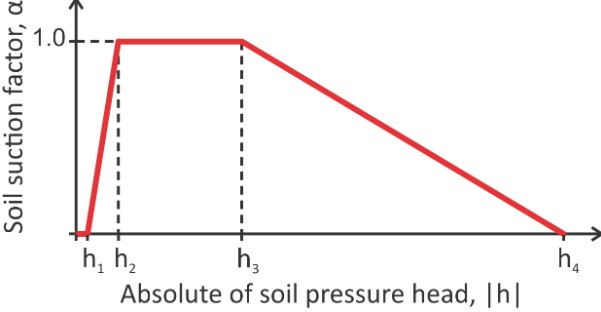

**Figure 4.** The general form of soil suction factor (adapted from [70]).

The second factor is the root density factor. Some researches (e.g., [71,72]) stated that the root length density declines exponentially with the depth and distance. Fatahi et al. [73] assumed that the maximum density of active roots (the fine non-woody roots which absorb water) is located at the distance of r = $r_0$ and the depth of z = $z_0$ and decreases exponentially in both radial and vertical directions (Figure 5). Therefore, the root length density can be expressed by [73]:

$$\beta = \beta_{\max} \cdot \exp\left(-k_1|z - z_0| - k_2|r - r_0|\right) \tag{6}$$

where $\beta_{\max}$ (L. L$^{-3}$) is the maximum root density and $k_1$ (L$^{-1}$) and $k_2$ (L$^{-1}$) are two experimental coefficients. Then Fatahi, Khabbaz, and Indraratna [73] presented the following hyperbolic tangent function for the root density factor:

$$G(\beta) = \frac{\tanh(k_3\beta)}{\int\limits_V \tanh(k_3\beta)dV} \tag{7}$$

where $k_3$ [L$^2$] is an empirical coefficient and V [L$^3$] is root zone volume.

The third term in Equation (4) is the potential transpiration factor. The potential transpiration is defined as the evaporation from plant leaves that depends on the meteorological condition of the atmosphere. In this study, the potential transpiration is given as input data. Because of the root resistance term, the water uptake is not constant in depth and the potential transpiration factor can be considered as linear [73]. Therefore:

$$F(T_P) = \frac{1 + k_4(z_{\max} - z)}{\int\limits_V G(\beta)(1 + k_4(z_{\max} - z))dV} T_P \tag{8}$$

where $k_4$ (L$^{-1}$) is an experimental coefficient and $z_{\max}$ (L) is the maximum depth of the root zone.

The root water uptake obtains with substituting three above factors (i.e., $\alpha(h)$, $G(\beta)$, and $F(T_P)$) in Equation (4). This value is calculated for each cell in the root zone.

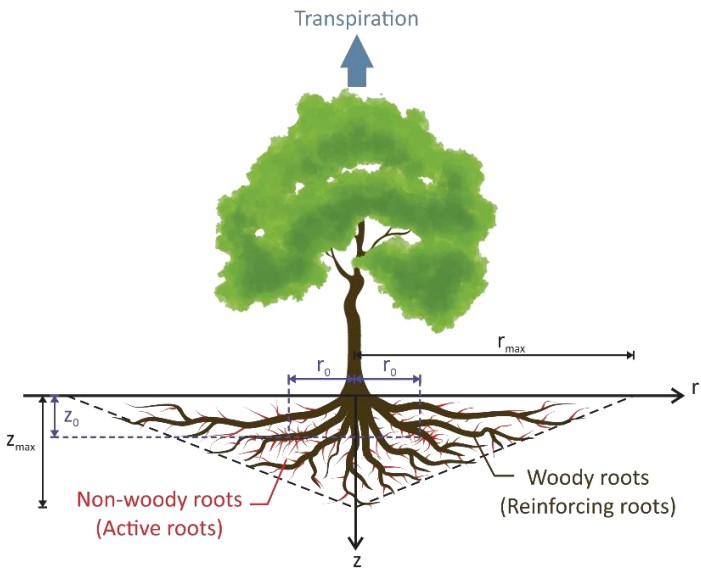

**Figure 5.** Schematic image of root distribution (modified from [66]).

### 3.1.4. Surcharge Load

The weight of vegetation is applied to the slope as a surcharge load and it is significant when the vegetation cover is heavy (e.g., large trees). This load increases the weight of slip mass (driving force) and also increases effective soil stress leading to higher shear strength (resistance force). Therefore, the surcharge load has both the stabilizing and destabilizing effects on the slope. The surcharge load is defined as vertical compressive stress applied on the ground surface. The size, density, and species of vegetation affect the surcharge due to the weight of vegetation [17]. Table 3 includes surcharge values reported by some researchers.

**Table 3.** The surcharge value due to the weight of trees.

| Species | Surcharge (kPa) | Size Indicator | Density (tree/ha) | Reference |
|---|---|---|---|---|
| *Sitka spruce* | Estimated: 5.2 (Used: 3.8) | Avg. height = 6 m | Dense | Wu et al. [19] |
| *Sitka spruce* | Average: 2.5 | | | Bishop and Stevens [74] |
| *Maritime pine* | 0.6 | | 350 | Kokutse et al. [75] |
| *Conifer forest* | up to 2 | Height = up to 80 m | Fully stocked | Greenway [17] |
| *Pinus sylvestris* | 3.5 | | | Van Asch et al. [76] |
| Riparian vegetation | 1.2 | Avg. height = 18 m | | Simon and Collison [77] |
| *Silver wattle* | Average tree: 0.81 Large tree: 5.06 | | 5000 | Abernethy and Rutherfurd [78] |
| *Pine* | 0.228, 0.135 | Age = 52-month | | Waldron and Dakessian [79] |
| *Korean pine* | 2.94 | Age = 20-year-old | Fully stocked | Kim et al. [80] |
| *European beech* | 0.309 | Avg. stem dia. = 14–42 cm | 308–2451 | Chiaradia et al. [81] |
| *Sweet chestnut* | 0.070 | Avg. stem dia. = 13–31 cm | 2268–3764 | |
| *Norway spruce* | 0.275 | Avg. stem dia. = 22–46 cm | 416–2066 | |
| *Mixed conifer forest* | 0.275 | Height ≈ 30 m | | Bischetti et al. [82] |

### 3.1.5. Effective Rainfall

The effective rainfall is the portion of rainfall that reaches the ground. To calculate effective rainfall, interception is subtracted from the total rainfall. Interception loss occurs due to the retention of water on canopies and trunks of vegetation and evaporation from these. Three main parameters influence interception loss, which are rainfall characteristics, vegetation structure, and meteorological conditions.

The simplest idea to calculate interception loss is a regression equation that relates gross rainfall to interception losses. Horton [83] introduces it for the first time as presented below.

$$I = a + bP \tag{9}$$

Here, $a$ (L) and $b$ (-) are constants related to vegetation type, and $P$ (L) and $I$ (L) are precipitation and interception loss respectively. In other words, it can be said the interception loss is a percentage of the precipitation in heavy rainfalls [83]. One of the advantages of this equation is the low number of parameters needed to define it.

Muzylo et al. [84] reviewed more realistic interception loss models of tall plants and found 15 models (10 original models and five improved models). Among these models, Rutter models (original and sparse) and Gash models (original and sparse) were the most commonly used formulas. However, the bulk of other models also are based on these. It is notable that in addition to the abundance of applications, the mentioned models have good validation [84].

The sparse Rutter model [27] and the sparse Gash model [28] are available in the effective rainfall module. In the sparse Rutter model, the amount of water reaching the ground surface can be calculated by:

$$P_{eff} = (1 - c)\int R \mathrm{d}t + c\left(\int D_{i,c}\mathrm{d}t + \int D_{t,c}\mathrm{d}t\right) \tag{10}$$

where $c$ (-) is the covered area ratio (the covered area divided by the total area), $R$ (LT$^{-1}$) is the gross rainfall rate, $D_{i,c}$ (LT$^{-1}$), and $D_{t,c}$ (LT$^{-1}$) are the drip from the canopy and the trunk drainage respectively. The value of $D_{i,c}$ is zero if the canopy is not saturated. After the saturation of the canopy, the excess water on the canopy storage capacity drains and a part of it ($P_d$) is inputted to trunk. Similar to the canopy, when the trunk is saturated, the excess water flows as trunk drainage ($D_{t,c}$). Here, $P_{eff}$ (L) is the effective precipitation which is the rainfall minus the interception loss.

The Gash sparse model is similar to the Rutter sparse model and it is simplified in some features. Gash et al. [28] formulated the amount of gross rainfall needed to saturate the canopy ($P'_g$) and trunks ($P''_g$). Accordingly, if the gross rainfall ($P_g$) is less than $P'_g$ or $P''_g$, the total of rainfall intercepts, the excess of gross rainfall is turned into throughfall or stemflow. These parameters are calculated with the following formula:

$$P'_g = -\frac{\overline{R}}{(1-\varepsilon)\overline{E}_c}\frac{S}{c}\ln\left[1 - \frac{(1-\varepsilon)\overline{E}_c}{\overline{R}}\right] \tag{11}$$

$$P''_g = \frac{\overline{R}}{\overline{R} - (1-\varepsilon)\overline{E}_c}\frac{S_t}{p_d c} + P'_g \tag{12}$$

Here, $\overline{R}$ (LT$^{-1}$) is average rainfall, $\overline{E}_c$ (LT$^{-1}$) is the average evaporation from the saturated covered area, and $\varepsilon$ (-) is the ratio of evaporation from saturated trunk to covered evaporation. Based on the definition of $\varepsilon$, the term of $(1-\varepsilon)\overline{E}_c$ shows evaporation from the saturated canopy and the term of $\varepsilon\overline{E}_c$ demonstrates the evaporation from the saturated trunks. The parameters $S$ (L) and $S_t$ (L) are the canopy and the trunk storage capacities, respectively. Other parameters were defined before. Gash et al. [28] presented the following equations to estimate throughfall and stemflow.

$$T_f = (1-c)\sum_{j=1}^{m+n} P_{g,j} + (1-p_d)c\left[1 - \frac{(1-\varepsilon)\overline{E}_c}{\overline{R}}\right]\sum_{j=1}^{n}\left(P_{g,j} - P'_g\right) \tag{13}$$

$$S_f = p_d c \left[ 1 - \frac{(1-\varepsilon)\overline{E}_c}{\overline{R}} \right] \sum_{j=1}^{q} \left( P_{g,j} - P'_g \right) - q S_t \tag{14}$$

where $m$ is the number of small storms which are not enough to saturate the canopy ($P_{g,j} < P'_g$), $n$ is the number of storms which are large enough to saturate the canopy ($P_{g,j} \geq P'_g$), $n + m$ is the total number of storms, and $q$ is the number of storms which are sufficient to saturate the trunks ($P_{g,j} \geq P''_g$). The summation of throughfall and stemflow can be used to obtain the available water on the ground surface.

### 3.1.6. Wind Force

When a single tree is exposed to the wind, it experiences a drag force which can be estimated by the following equation.

$$F_D = \frac{1}{2} C_D \cdot \rho_a V_w^2 \cdot A_t \tag{15}$$

where $F_D$ is the drag force (MLT$^{-2}$), $C_D$ is the drag coefficient (empirical) (-), $\rho_a$ is the air density (ML$^{-3}$), $V_w$ is the average wind velocity (LT$^{-1}$), and $A_t$ is the projected area of the tree against the wind. However, a forested slope is covered by many trees and the drag force should be calculated for all of them.

The wind force may increase the driving force of a slipping mass and decrease the stability of a slope while the wind direction is downslope. As noted earlier, considering downslope direction for the wind flow is a conservative assumption which is made in this study. When the vegetation cover is defined as a cluster of trees, it is easier to consider the wind effect on stability by Janbu's simplified method. Janbu's simplified method satisfies only the force equilibrium and having the sum of forces on trees is enough for a stability analysis. Whereas in the simplified Bishop method which considers the satisfaction of momentums, the location of trees and the location of drag force must be specified. Since detailed data usually is not available, in this version of the SSHV-2D model the wind effect is applied in Janbu's simplified method only.

### 3.2. Hydrological Module

The hydrology unit is included in the estimation of effective rainfall and the infiltration. Since the interception loss is related to vegetation cover, the infiltration of rainfall is covered in this module only.

The infiltration of effective rainfall to soil medium leads to a change in water content and matric suction in the unsaturated zone. As in Equation (19), the variation in matric suction influences the shear strength and thus changes the factor of safety. The Richards' equation [35] is the state of the art equation for water movement in unsaturated soil. The value of pressure head (or total head) in each point can be obtained by solving it, thereby leading to the matric suction (in the unsaturated zone) and pore water pressure (in the saturated zone). The two-dimensional Richards' equation can be written as:

$$\frac{\partial \theta}{\partial t} = \frac{\partial}{\partial x}\left( K_x(h) \frac{\partial h}{\partial x} \right) + \frac{\partial}{\partial z}\left( K_z(h) \frac{\partial h}{\partial z} - 1 \right) - S \tag{16}$$

where $\theta$ is volumetric water content (-), $K_x$ and $K_z$ are hydraulic conductivity (LT$^{-1}$) in the horizontal and vertical directions respectively, $h$ is pressure head (L), $z$ is the vertical dimension (L) which is positive downwards, and $S$ is the sink/source term (T$^{-1}$) for the root water uptake. Water content ($\theta$) is a function of pressure head ($h$) or vice versa. Equation (16) can be written in terms of pressure head alone (the $h$-based form), or in terms of water content (the $\theta$-based form), or in terms of both $\theta$ and $h$ (the mixed form). Because the $\theta$-based form only is valid in the unsaturated zone, it is not used in the current model. Celia et al. [85] demonstrated that the mixed form of Richards' equation is better at conserving the mass compared with the $h$-based form. Therefore, the mixed form is used in the current program.

The soil-water characteristic curve (SWCC) is a constitutive relationship between water content ($\theta$) and pressure head ($h$) in the unsaturated zone. Also, hydraulic conductivity ($K$) is a function of pressure head ($h$). Accordingly, Richards' equation is a nonlinear equation and it is impossible to find an analytical solution for general conditions, therefore, it is numerically solved.

Various empirical formulas exist for SWCC and the hydraulic conductivity function (HCF) [86–92]. In SSHV-2D, manual points can be entered by the user for SWCC and HCF. Several relationships are also available for selection (Table 4). These formulations are valid in both the unsaturated and saturated zones ($\theta = \theta_s$ and $K = K_s$). It has been demonstrated that the behavior of soil in wetting and drying cycles is not similar and the SWCC has a hysteresis property [93,94]. In the this stage of the SSHV-2D model, the hysteresis of the SWCC was not considered and wetting and drying cycles happen on a similar curve. The future development of the model this factor will be included.

**Table 4.** The available soil-water characteristic curve (SWCC) and hydraulic conductivity function (HCF) relationships in the integrated two-dimensional slope stability model (SSHV-2D).

| Reference | SWCC | HCF | Parameters |
|---|---|---|---|
| Haverkamp et al. [90] | $\theta(h) = (\theta_s - \theta_r)\dfrac{\alpha}{\alpha+|h|^{\beta}} + \theta_r$ | $K(h) = K_s\dfrac{A}{A+|h|^{B}}$ | $\alpha, \beta$: Fitting parameters <br> $A, B$: Fitting parameters |
| Van Genuchten [92] | $\theta(h) = \dfrac{\theta_s - \theta_r}{\left[1+(\alpha|h|)^n\right]^m} + \theta_r$ | $K(h) = K_s \dfrac{\left\{1 - \dfrac{(\alpha|h|)^{n-1}}{[1+(\alpha|h|)^n]^m}\right\}^2}{\left[1+(\alpha|h|)^n\right]^{\frac{m}{2}}}$ | $\alpha, n, m$: fitting parameters <br> where: <br> $m = 1 - \dfrac{1}{n}$ |
| Fredlund and Xing [87], Fredlund et al. [88] | $\theta(h) = C(\psi)\dfrac{\theta_s - \theta_r}{\left[\ln\left(e+\left(\frac{\psi}{a}\right)^n\right)\right]^m} + \theta_r$ <br> where: <br> $C(\psi) = 1 - \dfrac{\ln\left(1+\frac{\psi}{\psi_r}\right)}{\ln\left(1+\frac{10^6}{\psi_r}\right)}$ <br> $\psi = \gamma_w\cdot|h|$ | $K(h) = K_s\dfrac{\int_{\ln(\psi)}^{\ln(10^6)}\frac{\theta(e^y)-\theta(\psi)}{e^y}\theta'(e^y)dy}{\int_{\ln(\psi_{aev})}^{\ln(10^6)}\frac{\theta(e^y)-\theta_s}{e^y}\theta'(e^y)dy}$ <br> where: <br> $\theta'(\psi) = \dfrac{\partial\theta}{\partial\psi}$ | $e$: the natural number <br> $a, n, m$: fitting parameters <br> $\psi$: matric suction (varied between 0 to $10^6$ kPa) <br> $\psi_r$: matric suction corresponding to residual water content <br> $C(\psi)$: correction factor <br> $\psi_{aev}$: matric suction at air entry value <br> $y$: dummy variable of integration |

Notes: In all equations $\theta_s$ is saturated water content, $\theta_r$ is residual water content, $K_s$ is saturated hydraulic conductivity, and $h$ is pressure head ($h$ is negative in the unsaturated zone).

### 3.3. Slope Stability Analysis

The limit equilibrium approach is commonly used for slope stability analysis and it is used in the present study. The safety factor can be calculated by the single free-body methods and slice methods in the limit equilibrium approach. Single free-body procedures (e.g., Swedish circle, logarithmic spiral) only assume a slip surface and investigate the equilibrium for soil mass above it. Simplicity in use is the main advantage of these methods. In another category, vertical slices make the soil block above the slip surface. Slice methods are slightly more complex and include the following methods: ordinary method of slices, simplified Bishop, modified Bishop, simplified Janbu, Spencer, Morgenstern–Price, etc. These methods differ in slip surface type (circular or noncircular) and methods of considering interslice forces [95].

Here, we used the simplified Bishop [29] and Janbu's simplified methods [31,96] which are in the slice methods category. These methods need a low CPU time. Also, the safety factor obtained from the simplified Bishop method is close to the values from other methods [97]. Janbu's simplified method can work with a noncircular slip surface.

### 3.3.1. Shear Strength in the Saturated and Unsaturated Zone

In traditional analysis, the shear strength of the soil is expressed by the Mohr–Coulomb relationship as below. In the below equation, the effective normal stress ($\sigma'$ ($ML^{-1}T^{-2}$)) is replaced by the Terzaghi theorem [98].

$$\tau = c' + \sigma' \tan \phi' = c' + (\sigma - u_w) \tan \phi' \tag{17}$$

where $\tau$ is ultimate shear stress ($ML^{-1}T^{-2}$), $c'$ is the effective cohesion ($ML^{-1}T^{-2}$), $\sigma$ is total normal stress ($ML^{-1}T^{-2}$), $u_w$ is pore water pressure ($ML^{-1}T^{-2}$), and $\varphi'$ (-) is the effective internal friction angle. In this viewpoint, it is assumed that only pore water pressure is defined under the water table (in the saturated zone) but the pore water pressure is zero above the water table (in the unsaturated zone).

Bishop [33] proposed that the matric suction can be included in effective stress for the unsaturated zone:

$$\sigma' = \sigma - (u_a - \chi(u_a - u_w)) \tag{18}$$

where $u_a$ is the pore air pressure ($ML^{-1}T^{-2}$), $(u_a - u_w)$ is matric suction (positive) ($ML^{-1}T^{-2}$), and $\chi$ (-) is an empirical parameter called the matric suction coefficient. Bishop stated that the matric suction coefficient ($\chi$) is related to the degree of saturation and varies between 0 (for dry soil) and 1 (for fully saturated soil). In saturated conditions, $\chi = 1$ and the effective stress equation becomes $\sigma' = \sigma - u_w$, as suggested by Terzaghi [98]. Fredlund et al. [99] proposed an equation for shear strength of partially saturated soil:

$$\tau = c' + (\sigma - u_a) \tan \phi' + (u_a - u_w) \tan \phi^b \tag{19}$$

where $\varphi^b$ (-) is the friction angle relative to matric suction. By comparing Equation (19) with Equations (17) and (18):

$$\tan \phi^b = \chi \cdot \tan \phi' \tag{20}$$

This equation means that $\tan(\varphi^b)$ and $\chi$ have the same behavior. In many references, $\chi$ is correlated with degree of saturation (or matric suction) in the unsaturated zone (e.g., [33,100]) but in some references $\varphi^b$ has a constant value in the range of 0 to $\varphi'$ in unsaturated soil (e.g., [99,101]) and it is equal to $\varphi'$ in saturated medium.

In the SSHV-2D, two options are available for the effect of matric suction on shear strength. One is constant $\varphi^b$ and the other is $\chi = S_r$ (degree of saturation). In both cases, $\varphi^b = \varphi'$ when there is no matric suction (saturated zone). In the constant $\varphi^b$ case, if $\varphi^b = 0$ is used, the classic formulation for shear strength is used for slope stability analysis. Accordingly, the factor of safety equations was expressed in terms of shear strength in the unsaturated zone but it is also valid in the saturated zone.

### 3.3.2. Simplified Bishop Method

There are two main assumptions in the simplified Bishop method: (1) the slip surface is circular, and (2) interslice forces are horizontal, in other words, shear forces are ignored between slices. Figure 6 shows a slope with a circular slip surface. The factor of safety is obtained from the following equation.

$$SF = \frac{\sum \left[ \frac{((c'_s + c'_r) \cdot b + (W + q_{sur} \cdot b - u_a \cdot b) \tan \phi' + (u_a - u_w) \cdot b \tan \phi^b)}{m_\alpha} \right]}{\sum (W + q_{sur} \cdot b) \sin \alpha} \tag{21}$$

where

$$m_\alpha = \cos \alpha + \frac{\sin \alpha \tan \phi'}{SF} \tag{22}$$

Also, $c'_s$ ($ML^{-1}T^{-2}$) is effective cohesion of soil, $c'_r$ ($ML^{-1}T^{-2}$) is root contribution cohesion, $W$ ($MLT^{-2}$) is the weight of the slice, $q_{sur}$ ($ML^{-1}T^{-2}$) is the uniform surcharge load due to vegetation weight, $u_a$ ($ML^{-1}T^{-2}$) is pore air pressure (usually $u_a = 0$), $\varphi'$ (-) is the effective friction angle, $u_w$ ($ML^{-1}T^{-2}$) is pore water pressure, $u_a - u_w$ ($ML^{-1}T^{-2}$) is the matric suction, $\varphi^b$ is the angle indicating

the rate of change in shear strength relative to matric suction (in unsaturated zone $\phi^b \leq \varphi'$, and in saturated zone $\phi^b = \varphi'$), $b$ (L) is the width of the slice, $\alpha$ (-) is the inclination of slice base, and $SF$ (-) is the safety factor of the slope.

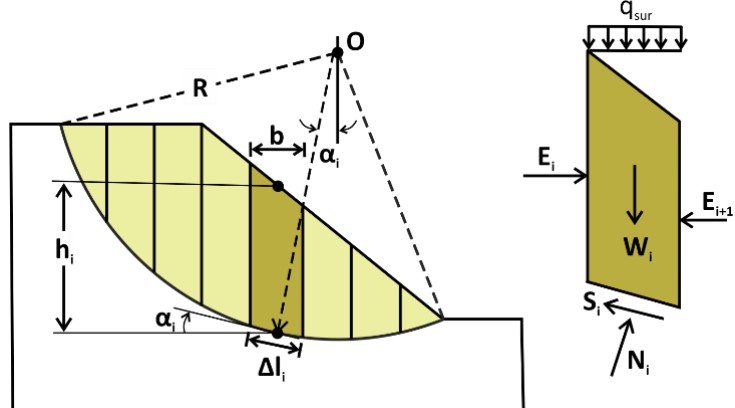

**Figure 6.** Circular slip surface and forces applied to a slice in the simplified Bishop method.

### 3.3.3. Janbu's Simplified Method

The slip surface in Janbu's simplified method [31] has a general form (circular or noncircular) as shown in Figure 7. This advantage allows modeling any surface to slip. A correction factor is applied to the factor of safety to account for the ignored interslice shear forces.

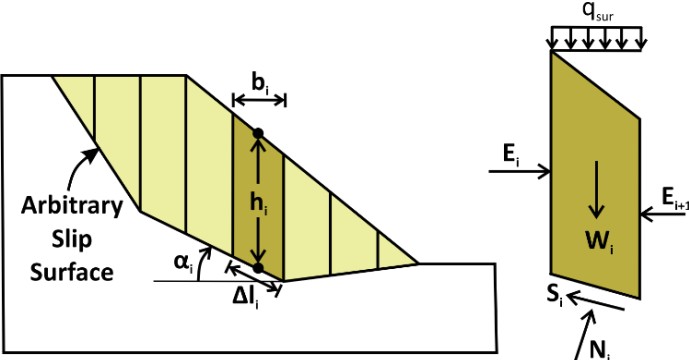

**Figure 7.** Noncircular slip surface and forces in Janbu's simplified method.

In this method, the factor of safety is calculated by:

$$SF = f_0 \cdot \frac{\sum \left( (c'_s + c'_r)\Delta l + (N - u_a \Delta l) \tan \phi' + (u_a - u_w)\Delta l \cdot \tan \phi' \right) \sec \alpha}{F_{wind} + \sum (W + q_{sur}b) \tan \alpha} \tag{23}$$

$$N = \frac{\left[ W + q_{sur}b - \frac{1}{SF}\left( c'_s + c'_r - u_a \tan \phi' + (u_a - u_w) \tan \phi^b \right)\Delta l \sin \alpha \right]}{m_\alpha} \tag{24}$$

where $\Delta l$ [L] is the length of the slice bottom, $N$ (MLT$^{-2}$) is a normal force on the base of the slice, $F_{wind}$ is the sum of drag forces acting to all trees, $f_0$ (-) is the correction factor, and $SF$ (-) is the factor of safety. Other parameters are similar to the parameters of the simplified Bishop method. Janbu et al. [31] presented a chart for estimating $f_0$ from the depth-to-length ratio of slipped mass (refer to Figure 8). Abramson et al. [102] presented the following formula for $f_0$ as a function of d and L which are defined in Figure 8.

$$f_0 = 1 + b_1 \left[ \frac{d}{L} - 1.4 \left( \frac{d}{L} \right)^2 \right] \tag{25}$$

wherein

$$b_1 = \begin{cases} 0.69 & \phi = 0 \text{ soils} \\ 0.31 & C = 0 \text{ soils} \\ 0.5 & C > 0, \phi > 0 \end{cases} \tag{26}$$

In SSHV-2D, the above equation is used for the estimation of $f_0$.

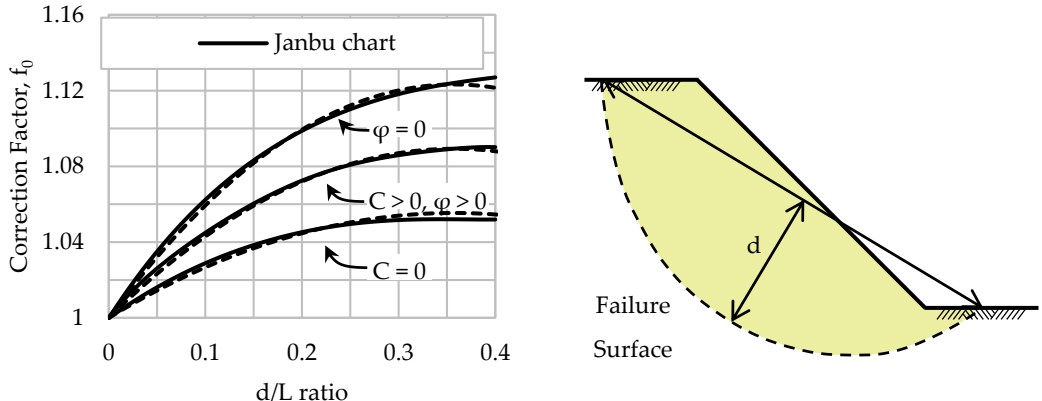

**Figure 8.** Correction factor for Janbu's simplified method [31,102].

## 4. Numerical Formulation

The numerical approach used in SSHV-2D is the finite difference method (FDM). In the following, the discretization of the domain, numerical formulation for hydrological unit and main numerical processes are described.

### 4.1. Geometry Discretization

The geometry of slope is defined by corner points. As shown in Figure 9a, at least six points are required to define the geometry of a slope. The number of required points is determined by the user. Also, the initial level of the water table can be defined by at least two points. After inputting the slope and water table points, the node spacing in the horizontal and vertical direction ($\Delta x$ and $\Delta z$) are determined. In FDM, the domain is covered by rectangular cells. According to meshing size ($\Delta x$ and $\Delta z$), the program discretizes the defined medium as shown in Figure 9b, converting the inclined boundaries into squared corners.

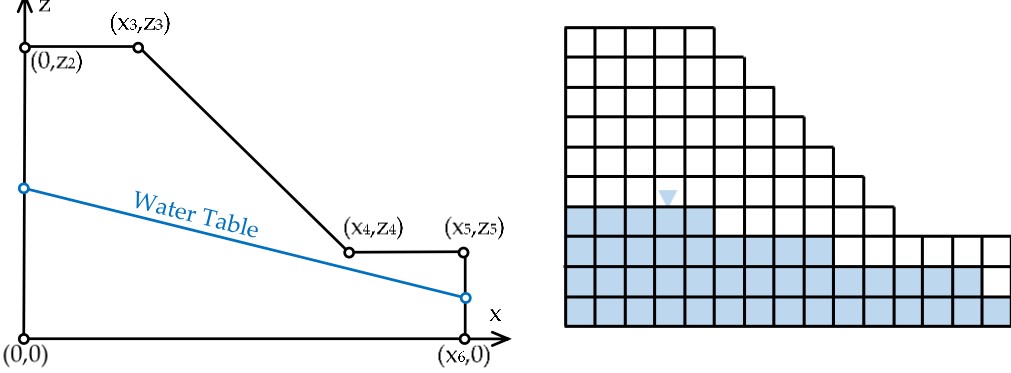

**Figure 9.** (**a**) The required nodes to define slope geometry; (**b**) the discretized domain in the finite difference method (FDM).

*4.2. Transient Water Infiltration*

**Discretization of equation:** The two-dimensional Richards' equation in mixed form is the governing equation in the hydrology module. To discretization this equation in time, explicit, fully implicit, and Crank–Nicolson schemes are available. Shahraiyni and Ataie-Ashtiani [103] demonstrated that the fully implicit scheme is more stable compared with Crank–Nicolson and two-stage Runge–Kutta (predictor–corrector) schemes. Therefore, the fully implicit scheme is used in the next problems and is described in the following. For spatial discretization, the centered scheme is selected because of higher-order accuracy. The Richards' equation is a nonlinear and must be solved by an iteration scheme. The Picard iteration method is used in SSHV-2D. The discretized form of Richards' equation for the node of $(i,j)$ can be written as below.

$$\left(\frac{\theta^{n+1,m+1} - \theta^n}{\Delta t}\right)_{i,j} = \frac{\Delta}{\Delta x}\left(k_x^{n+1,m}\left(\frac{\partial h}{\partial x}\right)^{n+1,m+1}\right)_{i,j} + \frac{\Delta}{\Delta z}\left(k_z^{n+1,m}\left(\frac{\partial h}{\partial z} - 1\right)^{n+1,m+1}\right)_{i,j} \tag{27}$$

Here, the subscripts of $i$ and $j$ show the spatial position of the node. The superscript of n or $n + 1$ demonstrates the current or next time step respectively, and the superscript m or $m + 1$ indicates the current or next iteration stage. Celia et al. [85] suggested using the following Taylor series for $\theta^{n+1,m+1}$:

$$\theta^{n+1,m+1} = \theta^{n+1,m} + \left(\frac{\partial \theta}{\partial h}\right)^{n+1,m}\left(h^{n+1,m+1} - h^{n+1,m}\right) + O\left(h^2\right) \tag{28}$$

Here $\frac{\partial \theta}{\partial h}$ is shown by $C$ $(\mathrm{T}^{-1})$ and is called the specific moisture capacity. If all nonlinear terms are ignored in the above equation, substituting in Equation (27) results in:

$$\frac{\theta_{i,j}^{n+1,m} + C_{i,j}^{n+1,m}\left(h_{i,j}^{n+1,m+1} - h_{i,j}^{n+1,m}\right) - \theta_{i,j}^n}{\Delta t} =$$
$$\frac{1}{\Delta x}\left((k_x)_{i+\frac{1}{2},j}^{n+1,m}\left(\frac{h_{i+1,j}^{n+1,m+1} - h_{i,j}^{n+1,m+1}}{\Delta x}\right) - (k_x)_{i-\frac{1}{2},j}^{n+1,m}\left(\frac{h_{i,j}^{n+1,m+1} - h_{i-1,j}^{n+1,m+1}}{\Delta x}\right)\right)$$
$$+ \frac{1}{\Delta z}\left((k_z)_{i,j+\frac{1}{2}}^{n+1,m}\left(\frac{h_{i,j+1}^{n+1,m+1} - h_{i,j}^{n+1,m+1}}{\Delta z} - 1\right) - (k_z)_{i,j-\frac{1}{2}}^{n+1,m}\left(\frac{h_{i,j}^{n+1,m+1} - h_{i,j-1}^{n+1,m+1}}{\Delta z} - 1\right)\right) \tag{29}$$

This relationship can be arranged as:

$$Ah_{i-1,j}^{n+1,m+1} + Bh_{i,j-1}^{n+1,m+1} + Dh_{i,j}^{n+1,m+1} + Eh_{i,j+1}^{n+1,m+1} + Fh_{i+1,j}^{n+1,m+1} = G \tag{30}$$

where:

$$A = \frac{1}{(\Delta x)^2}K_{ratio}\cdot K_{i-\frac{1}{2},j}^{n+1,m} \quad B = \frac{1}{(\Delta z)^2}K_{i,j-\frac{1}{2}}^{n+1,m}$$

$$E = \frac{1}{(\Delta x)^2}K_{ratio}\cdot K_{i+\frac{1}{2},j}^{n+1,m} \quad F = \frac{1}{(\Delta z)^2}K_{i,j+\frac{1}{2}}^{n+1,m}$$

$$D = -\frac{C_{i,j}^{n+1,m}}{\Delta t} - A - B - E - F$$

$$G = \frac{\theta_{i,j}^{n+1,m} - \theta_{i,j}^n}{\Delta t} - \frac{C_{i,j}^{n+1,m}}{\Delta t}h_{i,j}^{n+1,m} - \frac{1}{\Delta z}K_{i,j-\frac{1}{2}}^{n+1,m} + \frac{1}{\Delta z}K_{i,j+\frac{1}{2}}^{n+1,m} + S_{i,j}$$

$h_{i,j}^{n+1,m+1}$ is the pressure head in the node $(i,j)$ in time step $n + 1$ and iteration stage $m + 1$. Also, $\Delta t$ (T) is time increment, $K$ $(\mathrm{LT}^{-1})$ is the hydraulic conductivity in the vertical direction, and $K_{ratio}$ (-) is the ratio of hydraulic conductivity in the horizontal direction to the vertical direction $(= K_x/K_z)$. Other parameters were already introduced.

Applying Equation (30) to all nodes leads to a linear system of equations as $[A]\{h\} = \{B\}$. The dimension of the system is equal to the number of nodes and the solution of this is the values of pressure head in iteration $m + 1$. When the difference between values of pressure head in iteration m and $m + 1$ is less than the allowed tolerance, the values of pressure head in time step $n + 1$ are obtained.

In the mentioned system of equations, the coefficient matrix, (A), is a bounded and symmetric matrix which has five nonzero elements in each row. The Gaussian elimination method is used to solve this system. Since the soil pressure head (the variable of the above system of equations) is valid in both unsaturated and saturated zones, the system of equations can be implemented to whole nodes of the domain.

**Averaging of hydraulic conductivity:** In the above formulation, the hydraulic conductivity terms are defined as an average of two adjacent nodes. For example, $K_{i-\frac{1}{2},j}^{n+1,m}$ is the average of $K_{i-1,j}^{n+1,m}$ and $K_{i,j}^{n+1,m}$. Three averaging approaches are considered in SSHV-2D: arithmetic averaging, geometric averaging, and harmonic averaging [104]. Table 5 shows averaging relationships for each method. Since Richards' equation is a nonlinear equation, it seems that the non-arithmetic averaging can increase the rate of convergence.

**Table 5.** The averaging methods for hydraulic conductivity in SSHV-2D.

| Parameter | Arithmetic Method | Geometric Method | Harmonic Method |
|:---:|:---:|:---:|:---:|
| $K_{avg}$ | $\frac{1}{2}(K_1 + K_2)$ | $\sqrt{K_1 \times K_2}$ | $\frac{K_1 \times K_2}{K_1 + K_2}$ |

**Specific moisture capacity:** Another important parameter in Richards' equation is specific moisture capacity ($C = \frac{\partial \theta}{\partial h}$). For the evaluation of $C$ in any node, the pressure head is substituted in the first-order derivative of the SWCC relationship. This scheme is called tangent approximation. Rathfelder and Abriola [105] proposed an alternative named standard chord slope (SCS) that is expressed by Equation (15). They demonstrated that the SCS scheme is better at mass conservation than the tangent approximation in the h-based Richards' equation. Both schemes for evaluation of $C$ are available in the SSHV-2D program.

$$C_{i,j}^{n+1,m} = \frac{\theta_{i,j}^{n+1,m} - \theta_{i,j}^n}{h_{i,j}^{n+1,m} - h_{i,j}^n} \tag{31}$$

**Time stepping schemes:** The infiltration of rainfall in the soil is a transient problem and time stepping value ($\Delta t$) can be effective in achieving a faster solution to the problem. In the program, in addition to constant time-stepping, automatic time stepping [103,106] is also available which adaptively determines $\Delta t$ in the next step.

$$\Delta t^{new} = \begin{cases} 1.3\,\Delta t^{old} & N \leq 3 \\ \Delta t^{old} & 3 < N \leq 7 \\ 0.6\,\Delta t^{old} & 7 < N \leq N_{max} \end{cases} \tag{32}$$

Here, $N$ is the number of iterations in each step, $N_{max}$ is allowed maximum number of iterations, and $\Delta t^{new}$ (T) and $\Delta t^{old}$ (T) are time steps of the next and current step respectively. If the number of iterations ($N$) was greater than $N_{max}$, then $\Delta t$ is converted to $\Delta t/3$ and the current iteration is repeated again. In automatic time-stepping, the time increment is increased if the solution condition is good (the number of iterations is low) and decreases if the solution condition is bad (the number of iterations is high).

**Boundary conditions (BC):** For boundaries of the domain, two main conditions are considered in the SSHV-2D program: constant head (Dirichlet BC) and flux (Neumann BC). In the former, the value of the

head (unknown of the problem) in boundary nodes is definite and given by the user. But, in the second case, the flux on the boundary is given and the rate of the head is definite. According to Darcy's law:

$$q = -k\frac{\partial H}{\partial n} = \begin{cases} -K_x\frac{\partial h}{\partial x} & \text{in horizontal direction} \\ -K_z\left(\frac{\partial h}{\partial z} - 1\right) & \text{in vertical direction} \end{cases} \tag{33}$$

This means that one of the terms in the right hand side of Equation (29) must be replaced with $-q$ according to the location of the boundary.

A special case of flux BC is intended for modeling of rainfall in the SSHV-2D program. In general, when the rainfall is greater than the infiltration capacity of the soil, a portion of rainfall cannot infiltrate the soil and remains on the ground surface. This water can turn into runoff and flow away, or stay in place (ponding). Here it is assumed that excess water turns into the runoff. In this condition, incoming flux is equal to rainfall rate while the pressure head at the ground surface is negative. If the pressure head at the ground surface became greater than zero, the boundary condition converted to constant head with $h = 0$ value set in place automatically.

### 4.3. Slope Stability Analysis

The factor of safety (*SF*) for a slope is obtained by Equation (21) or (23) according to the selected method. One of the required parameters in these equations is the width of slices ($b$) which in SSHV-2D is equal to the horizontal mesh size ($\Delta x$) to reduce the number of inputs. Other parameters ($C'$, $\varphi'$, $\varphi^b$, $W$, $u_w$, etc.) are related to soil and vegetation properties or the result of Richards' equation. The user should determine the position of the slip surface so that *SF* can be obtained. In the SSHV-2D program, the slip circles are defined by center and radius. The centers are located above the slope and the interval between, along with the range of radius changes, and the increment of the radius is determined by the user. Figure 10 demonstrates the position of centers and the radii for a slip circle. The critical *SF* is the minimum of *SF* for all defined circles which is determined by the program. In Janbu's method, the slip surface can be noncircular, and the noncircular failure surface is manually defined by the user.

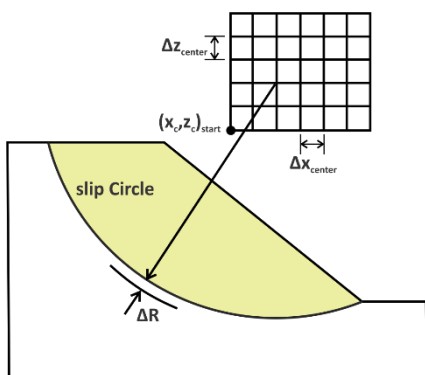

**Figure 10.** Centers and radii for slip circles in slope stability analysis.

## 5. Verifications and Results

In this section, some common benchmarks and experimental data which have been reported in the literature are used to verify program results. The final result of the current program is safety factor (*SF*) of the slope, but some problems are used for verification of individual modules of the program without regard to the *SF*.

### 5.1. Example 1: 1-D Infiltration in the Vertical Soil Column

Haverkamp et al. [90] investigated one-dimensional water movement in an unsaturated soil column. They compared experimental results of infiltration into a sand column with results of

different schemes of numerical methods. In this experiment, the soil column has a height of 93.5 cm and a diameter of 6 cm in a plexiglass casing that was equipped with tensiometers. As boundary condition, a constant flux $q$ = 13.69 cm/h was applied at the soil surface and a constant water content $\theta$ = 0.1 cm$^3$/cm$^3$ (or water pressure head of −61.5 cm) was maintained at the bottom of the column. The water content in the entire soil column was imposed $\theta$ = 0.1 cm$^3$/cm$^3$ at the start time.

The analysis of obtained data from the experiment led to hydraulic conductivity and water content relationships versus water pressure head as presented in Table 4. According to laboratory data $K_S$ = 34 cm/h, $\theta_S$ = 0.287, $\theta_r$ = 0.075 and by using the least-square fit on the data, the empirical coefficients $A$ = 1.175 × 10$^6$, $B$ = 4.74, $\alpha$ = 1.611 × 10$^6$, and $\beta$ = 3.96 were determined. In both equations, $h$ is water pressure head in cm.

The soil column was modeled in SSHV-2D program by the above parameters. The spatial increment $\Delta z$ = 1 cm and the temporal increment $\Delta t$ = 0.001 hr were used for simulation. Figure 11 shows numerical results obtained from SSHV-2D by experimental results presented in the original reference. It is noticeable that there is good agreement between numerical results and laboratory data. The numerical scheme to solve Richards' equation was assumed backward in time (implicit scheme) and centered in space.

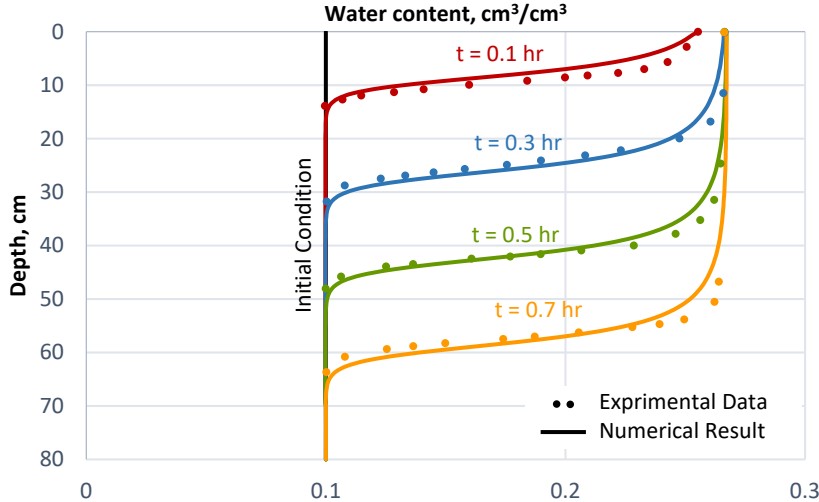

**Figure 11.** The result of hydrology module for one-dimensional infiltration problem.

## 5.2. Example 2: 2-Dimensional Infiltration and Water Table Recharge

Vauclin et al. [107] investigated transient 2-D water flow for shallow free aquifers. For this purpose, they assumed a slab of homogeneous soil, 6 m in length and 2 m in height, ending in trenches from the right and left sides. The water elevation in trenches was maintained at 65 cm (at the depth of 135 cm from the soil surface) which led to a horizontal water table at the depth of 135 cm in the soil. In the top of the medium, the water with a constant rate of 14.8 cm/h was applied to the soil at the center of the trenches with a length of 1 m, and evaporation was stopped on the remaining parts of the soil surface. Due to the symmetry of the problem, in the original reference, only one-half of the medium was modeled in the laboratory and the recharge of the water table was measured at different times.

Figure 12 shows the right half of the medium with dimensions of 300 cm × 200 cm. The left boundary condition was no-flow due to symmetry. In the right boundary, the water level was fixed in the trench; and above the water, a no-flow boundary condition was imposed. The bottom of the model was impervious and the soil surface was sealed against evaporation except for the portion of the surface where infiltration occurred (50 cm from top left).

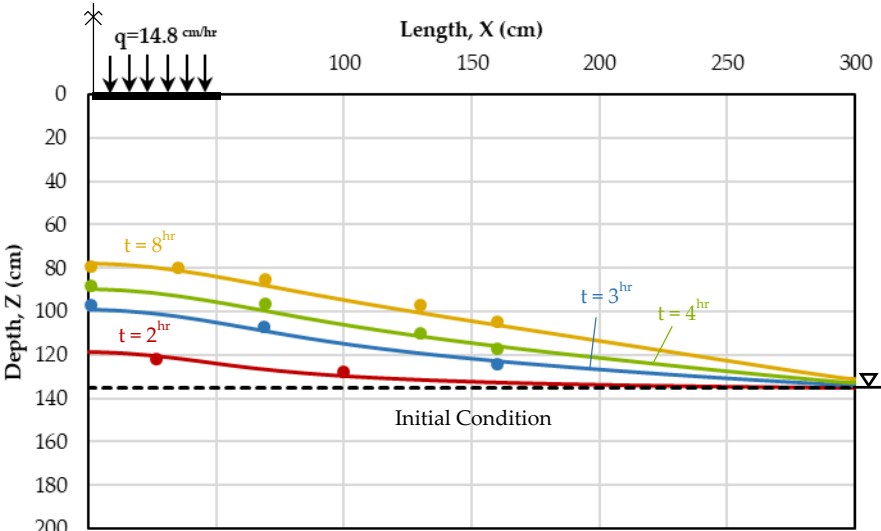

**Figure 12.** Result of 2-D water table recharge example.

In modeling the current benchmark by SSHV-2D, the boundary condition was defined as considered above and the initial condition was assumed to be hydrostatic for the initial water table position. Haverkamp's model was used to define the unsaturated soil constitutive relationships (SWCC and hydraulic conductivity function). The fitting parameters for these relations were determined by the least-squares method from experimental data by Vauclin et al. [107]. The required parameters in modeling can be found in Table 6.

**Table 6.** Summary of parameters used in the water table recharge problem [107].

| Parameter | Value |
|---|---|
| Soil-Water Characteristic Curve (SWCC) parameters | |
| $\theta_S$ (cm³/cm³) | 0.30 |
| $\theta_r$ (cm³/cm³) | 0 |
| $\alpha$ | 40,000 |
| $\beta$ | 2.90 |
| Hydraulic conductivity function parameters | |
| $K_s$ (cm/h) (in both horizontal and vertical directions) | 35 |
| A | $2.99 \times 10^6$ |
| B | 5.0 |
| Numerical assumption | |
| $\Delta x$ (cm) | 10 |
| $\Delta z$ (cm) | 5 |
| $\Delta t$ (hr) | 0.1 |
| Time stepping method | Automatic |
| Hydraulic conductivity averaging method | Geometric |
| Estimation method of specific moisture capacity | SCS |

The numerical results and experimental data are shown in Figure 12. A good correlation between numerical and experimental results demonstrates that the transient water movement module of the model works well. Clement et al. [108] investigated this problem numerically. They used FDM for numerical modeling of this problem. Their methods were similar to what used in this study. The only difference was that they used the arithmetic method for averaging the hydraulic conductivity. The comparison between results demonstrates that the geometric method gives a result closer to experimental data.

*5.3. Example 3: Stability Analysis of a Slope*

**Case 1—A homogenous slope:** A homogeneous slope which was used in the studies of Fredlund and Krahn [97] and Xing [109] is shown in Figure 13. The properties of soil in this example are $\gamma$ = 19.2 kN/m$^3$, $C'$ = 29.3 kPa, and $\varphi'$ = 20° (the values used in various references are slightly different). They reported safety factors of the slope for different methods. Other researchers applied their proposed methods on this problem and reported the safety factor of the slope (Table 7).

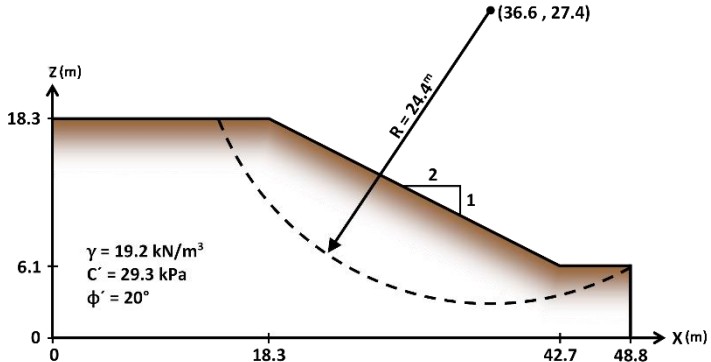

**Figure 13.** Homogeneous slope studied by Fredlund and Krahn [97].

**Table 7.** Reported safety factors for homogeneous slope (Case 1).

| Reference | Analysis Method | SF |
|---|---|---|
| Fredlund and Krahn [97] | Simplified Bishop Method | 2.080 |
| | Janbu's Simplified Method | 2.041 |
| Xing [109] | Proposed 3D Method | 2.122 |
| Chen et al. [110] | Upper bound method | 2.262 |
| Chen et al. [111] | Proposed 3D Method (STAB-3D) | 2.188 |
| | Plain-Strain 3D Method | 2.073 |
| Sultan et al. [112] | Upper Bound Theorem (SAMU-3D Program) | 2.213 |
| Ge [113] | Vector Sum Method (VSM) | 2.037 |
| Sun et al. [114] | Proposed 3D Method | 2.000 |
| Liu et al. [115] | 3D independent cover-based manifold method (ICMM3D) and vector sum method (VSM). | 2.061 |
| This Study (SSHV-2D) | Simplified Bishop Method | 2.079 |
| | Janbu's Simplified Method | 2.024 |

The mentioned slope was modeled by the SSHV-2D program. Assuming $\Delta x = \Delta z = 0.1$ m as mesh dimensions, the *SF* = 2.079 and 2.024 were obtained for Bishop and Janbu's simplified methods respectively. Xing [109] stated the center of the slip circle is (36.6 m, 27.4 m) and its radius is 24.4 m as shown in Figure 13. The safety factor was calculated for this center and radius, but our program gave lower safety factors occurring for another center point and radius, with *SF* = 2.003 and 1.970 for Bishop's and Janbu's methods, respectively.

**Case 2—A homogenous slope with a weak layer:** After the analysis of the circular slip surface, Fredlund and Krahn [97] and Xing [109] added a weak layer to the slope as shown in Figure 14. This layer was so weak so that it changed the slip surface from circular to composite slip surface. The different researchers computed the *SF* of this problem for a given center point and radius, but the position of the weak layer was different in these references. Accordingly, the elevation of the weak

layer from the bottom of the model is defined by D (see Figure 14). Here, we used Janbu's simplified method and calculated SFs for the determined center point and radius which are compared with other researcher's results in Table 8. It is notable that, similar to the first case, the critical slip circle is identified different from the given circle, and the minimum safety factors are 1.366, 1.413, and 1.434 for D = 6.1 (oblique), 4 and 4.55 m, which is slightly different from values in Table 8.

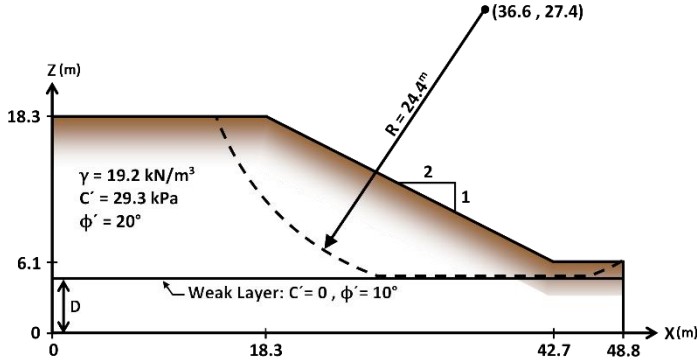

**Figure 14.** Homogeneous slope with a weak layer, studied by Fredlund and Krahn [97].

**Table 8.** Reported safety factors for the homogeneous slope with a weak layer (case 2).

| Reference | Analysis Method (Program Name) | D (m) | Condition of Weak Layer | SF |
|---|---|---|---|---|
| Fredlund and Krahn [97] | Simplified Bishop Method | 6.1 | Oblique * | 1.377 |
|  | Janbu's Simplified Method | 6.1 | Oblique * | 1.448 |
| Li and White [116] | New Proposed Method | 4 | Horizontal | 1.387 |
| Xing [109] | Proposed 3D Method | 6.1 | Oblique * | 1.548 |
| Hungr et al. [117] | 3D extension of the Bishop's Simplified method (CLARA) | N/A | N/A | 1.62 |
| Lam and Fredlund [118] | Janbu's Simplified Method | 5 | Horizontal | 1.558 |
| Huang and Tsai [119] | Modified Bishop Simplified Method | ~4.6 | Horizontal | 1.658 |
| Kim et al. [120] | Lower-Bound Method | 4.6 | Horizontal | 1.25 |
|  | Upper-Bound Method |  |  | 1.37 |
| Chen et al. [111] | Proposed 3D Method (STAB-3D) | ~4.6 | Horizontal | 1.64 |
|  | Plain-Strain 3D Method |  |  | 1.384 |
| Ge [113] | Vector Sum Method (VSM) | 4.55 | Horizontal | 1.585 |
| Zheng [121] | Proposed 3D Method | 4.55 | Horizontal | 1.707 |
| Sun et al. [114] | Proposed 3D Limit Equilibrium Method | 4.55 | Horizontal | 1.68 |
| Zheng [122] | Spencer's Method (RMP3D) | 4.55 | Horizontal | 1.735 |
|  | Corps of Engineers Assumption |  |  | 1.766 |
| Liu et al. [115] | 3D independent cover-based manifold method (ICMM3D) and vector sum method (VSM). | 4.55 | Horizontal | 1.530 |
| This Study | Janbu's Simplified Method (SSHV-2D) | 6.1 | Oblique * | 1.391 |
|  |  | 5 | Horizontal | 1.446 |
|  |  | 4.55 | Horizontal | 1.489 |

\* $\theta = -1°$.

A comparison between obtained results in the current study with others demonstrates that SSHV-2D works well. Relative to reference results [97], calculated SFs in case 1 have a deviation of 1% and the deviation in case 2 is below 4%.

### 5.4. Example 4: The Effect of Matric Suction on the Stability of a Slope

Griffiths and Lu [123] investigated the stability of slopes by considering unsaturated soil and matric suction effect. They used the Slope64 program which works by the finite element method. One of the cases presented in their paper is a homogeneous slope with different levels of the water table as shown in Figure 15a. The properties of soil are $\gamma = 20$ kN/m$^3$, $C' = 10$ kPa, and $\varphi' = 20°$. To investigate the matric suction effect, two conditions of pore pressure were considered. In the first condition, the matric suction was ignored and in the other case, stability was analyzed by considering matric suction.

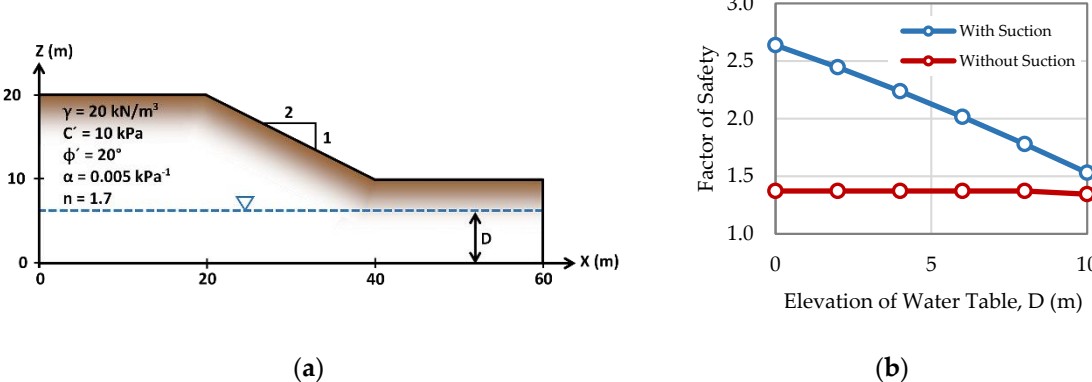

(**a**)  (**b**)

**Figure 15.** (**a**) The geometry of the slope, and (**b**) safety factor (*SF*) for different elevations of the water table.

Griffiths and Lu [123] considered the increase in shear strength of soil due to matric suction by Equation (19). They defined $\phi^b$ as a function of the effective degree of saturation (or matric suction) as Equation (20). The matric suction coefficient ($\chi$) with the Van-Genuchten model can be written as:

$$\chi = \frac{S - S_r}{1 - S_r} = \left( \frac{1}{1 + (\alpha(u_a - u_w))^n} \right)^{1 - \frac{1}{n}} \tag{34}$$

where $S$ is the degree of saturation and $S_r$ is the residual degree of saturation. The Van-Genuchten fitting parameters were assumed as $\alpha = 0.005$ kPa$^{-1}$ and $n = 1.7$ in the original reference.

Figure 15b shows the result of the SSHV-2D program for stability analysis of the slope. Although in both conditions the SFs are greater than 1.0 and the slope is stable, consideration of the matric suction effect improves the *SF* close to two times.

The obtained SFs are close to values reported by Griffiths and Lu [123]. The maximum difference between SFs is <3% which is because of the solving method and the assumptions. Good agreement between results demonstrates that the program works well for the unsaturated condition. In both studies, Bishop's method is used to analyze slope stability.

### 5.5. Example 5: Improvement of Slope Stability by Vegetation

In the current example, the impact of root reinforcement on slope stability is investigated. Zhu et al. [60] considered a slope with a height of 15 m and an angle of 37° which was covered with trees. As shown in Figure 16, the slope profile includes two layers: a very stiff soil as the bottom layer and completely decomposed granite (CDG) as the top layer. The thickness of the surface layer is D which changes from 1 to 4 m. The groundwater table is fixed and has a 1:6 inclination. This profile simulates a typical slope that was observed in a landslide in Hong Kong [60].

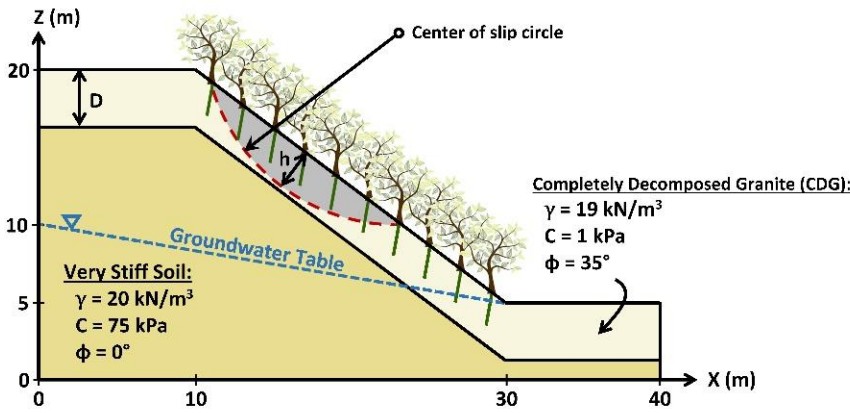

**Figure 16.** Schematic profile of slope used in example 5 [60].

In this example, the roots of trees are assumed to be a bar that extends deeply in the soil with a length of 2.2 m. The number of roots per unit length of the slope or root density (RD) was varied from 0 to 0.8 [60]. The enhancement of shear resistance due to root reinforcement is calculated by Equation (3). The root parameters used in the current example are listed in Table 9. According to these parameters, the threshold length of roots determines the governed resistance mechanism, which is 5 m. Because the threshold length is greater than root length, the dominant resistance mechanism is pullout resistance in all cases. It should be mentioned that in this example the surcharge due to trees has been ignored and other effects of vegetation have not caused concern, because the original authors were focused on root reinforcement as one of the mechanical effects of vegetation.

**Table 9.** Root properties for modeling of soil–root interactions [60].

| Definition | Parameter (Unit) | Value |
| --- | --- | --- |
| Root tensile capacity | T (kN) | 12.5 |
| Root pullout resistance | P (kN/m) | 2.5 |
| Lateral bending strength of the root | Q (kN) | 6.25 |

This slope was modeled by SSHV-2D and the impacts of root reinforcement and root density were investigated. We used mesh size of $\Delta x = \Delta z = 0.1$ m and the increments of the center and radius of slip circle were $\Delta x_{center} = \Delta z_{center} = \Delta R = 0.1$ m. The safety factor of the slope and the maximum depth of the slip surface (h in Figure 16) are presented in Figure 17. The results demonstrate that the increase in root density enhances the stability of the slope. For instance, when the surface layer thickness is D = 1 m, the safety factor increases from 1.13 (in bare slope) to 1.66 (in RD = 0.8 m$^{-1}$) which means 47% improvement in stability. Also, increasing the slip surface depth with root density indicates that the vegetation can make the surface layer more cohesive and prevent shallow landslides. Specifically, in the case that the thickness of the surface layer is low (D = 1 m) and the root density is high (RD > 0.4), the surface soil has become so strong that the slip surface is removed from the surface layer and is placed in a very stiff layer (see Figure 17b). In other thicknesses of the surface layer, this does not happen because the length of the roots is not sufficient to strengthen the whole depth of the surface layer.

The obtained results are close to the main reference results and the average of errors for safety factors is 2%, which is acceptable. The similarity between outputs of the program and reference results shows good performance of the program.

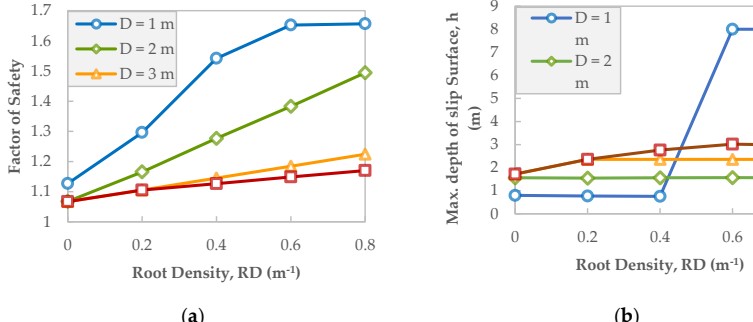

**Figure 17.** (**a**) The factor of safety and (**b**) the maximum depth of the slip surface for the vegetated slope.

## 6. Conclusions

Integrated impacts of hydrology and vegetation on slope stability were investigated in this work. The SSHV-2D model as is a platform for further studies was developed. Model results were compared against various benchmark problems and reported experimental data which showed that different modules of the model have satisfactory performance. In these problems, it was found that consideration of matric suction in the unsaturated zone improves the stability of the slope more than 90%. This means that the infiltration of rainfall which increases the soil water content and consequently decreases the matric suction of the soil reduces the stability of slopes. Moreover, it was demonstrated that the roots of trees can increase the safety factor of a slope, stabilize surface layers of the soil, and prevent shallow landslides. For instance, in the case of the vegetated slope which was mentioned in this paper, trees with a density of 0.8 m$^{-1}$ increased the safety factor from 1.13 (in bare slope) to 1.66, which means about 50% improvement of stability. Future works will be focused on sensitivity analysis of vegetation characteristics and the effect of hydrological parameters on the stability of slopes in real-world problems.

**Author Contributions:** M.E.-T.: model development, simulations and writing original draft preparation; B.A.-A.: formulation of the research questions, reviewing the paper and research supervision.

**Funding:** This research received no external funding

**Acknowledgments:** Behzad Ataie-Ashtiani acknowledges the support of the Research Office of the Sharif University of Technology, Iran and the National Centre for Groundwater Research and Training, Australia.

**Conflicts of Interest:** The authors declare no conflict of interest.

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
