# Peer review of "A Modeling Platform for Landslide Stability: A Hydrological Approach"

_water, doi:10.3390/w11102146_

Round 1
Reviewer 1 Report
Please see the attached file

Author Response
Point 1: A very important issue is concerned to material heterogeneity that has not been properly discussed, at least in the Introduction. The heterogeneity and, accordingly, the numerical distribution of the values of important parameters could have a not evanescent impact on the stability of the system under study and, accordingly, on the resulting Factor of Safety (locally as well) as discussed, among many others, in the following papers:
Vu-Bac N., Lahmer T., Zhuang X., Nguyen-Thoi T., Rabczuk T., 2016. A software framework for probabilistic sensitivity analysis for computationally expensive models. Adv. Eng. Softw. 100, 19 31. Pasculli A., Calista M., Sciarra N. (2018). Variability of local stress states resulting from the application of Monte Carlo and finite difference methods to the stability study of a selected slope. Engineering Geology. Vol. 245, pp. 370-389.
Response 1: Authors appreciate the supportive and constructive comments of the reviewer. This point has been considered at the end of section 2.1 and the references are considered and added.
Point 2: Hysteresis could have an important impact on the stability of unsaturated slope and on the material integrity, as discussed, respectively, in:
Ma, K.C., Tan, Y.C., Chen, C.H., 2011. The influence of water retention curve hysteresis on the stability of unsaturated soil slopes. Hydrol. Process. 25, 3563-3574. Pasculli, A., Sciarra, N., Esposito, L., Esposito, A.W., (2017). Effects of wetting and drying cycles on mechanical properties of pyroclastic soils. Vol. 156, pp. 113-123.Accordingly, a brief, related discussion should be addressed.
Response 2: This point has been mentioned at the above table 4. The references are considered and added. In the future development of this model the influence of hysteresis will be considered. However, it shall be noticed that the hysteresis influence is commonly a secondary factor compared to the considered factors in this study.
Point 3: Another important issue should be addressed through more details in the “Discretization of equations” section: the unsaturated-saturated and vice-versa transition, in particular by a numerical point of view, but also by hysteresis point of view. For example:
How equations 29, 30 and those reported in lines 480 and 481 are modified? Furthermore, how they change during the transient in saturated areas? Perhaps, a simple flow chart may help. Line 486: which was the selected algorithm for matrix inversion? Line 494: which was the criterion for the selection of the averaging method? Lines 499-500: Eq. 33 is the correct one?
Response 3:
Richards’ equation is a nonlinear equation because the coefficients of the equation are a function of equation variable. We used Picard’s iteration method to solve it. So that an initial value is considered for equation variable (the value of the variable in pervious time step is a good value to start) to calculate the coefficients of the equation, then with known coefficients the equation variable is computed. In the following, the new value of variable is considered to calculate equation coefficients. Again, the equation is solved with updated coefficients and the value of variable in new iteration stage is obtained. This process is repeated until the difference between the assumed value of variable to calculation of coefficients and the obtained value of equation variable is less than an accepted value. When this condition was established, the value of variable in the current time step is obtained and this process must be repeated for next time step. The coefficients of equation (A, B, D, E, and F: in below of eq. 30) is included hydraulic conductivity, mesh sizes, and specific moisture capacity. The mesh sizes are constant in both saturated and unsaturated zone. The hydraulic conductivity in unsaturated zone is a function of head and in saturated zone is constant (Ks). Also, specific moisture content is dependent on soil moisture in unsaturated zone and is equal to zero in saturated zone (because the soil moisture in saturated zone is always equal to θs). In other words, the coefficients of the equation are constant in saturated zone and don’t change with time. The Gaussian elimination method is used to solve the system of equation. This has been added to the manuscript. Since Richards’ equation is a nonlinear equation, it seems that the non-arithmetic averaging can increase the rate of convergence. Various methods of averaging have been implemented in the model. Yes, it is correct.
Point 4: For the simulations carried out, a numerical test for the verification of the optimal choice of the size of the element adopted and therefore of the number of elements considered was not reported. The aim is to check that, with the increase in the number of elements, the results do not change significantly.
Response 4: The test for independence of the results to the mesh size has been performed and the considered mesh sizes are suitable.
Point 5: Line 630: in the framework of comparison between numerical results (therefore not with analytical results), “deviations” and not “errors”.
Response 5: This is true and was corrected.
Point 6: Lines 653-683: the effect of the weight of the trees, also linked to their height, was considered only marginally. It is reasonable to expect that the beneficial effect of the roots on stability can be somehow balanced by the negative effect of tree loading. So it would be reasonable to expect a descending trend of safety factors to the increase of root density, proportional to the increase of tree density. Moreover, in fig. 17b, why around 0.4 in abscissa there is a steep increment of the Max depth of slip surface related to D= 1m, not occurred for the other values of D?
Response 6: The weight of the trees can have both a positive and a negative effect. This effect is dependent on the slope angle. It can be shown that increasing the slope angle can cause negative effects of the surcharge and decreasing it can improve the factor of safety. But the rate of decrease or increase in the factor of safety is not significant (according to below reference).
Kokutse, N. K., Temgoua, A. G. T., & Kavazović, Z. (2016). Slope stability and vegetation: Conceptual and numerical investigation of mechanical effects. Ecological Engineering, 86, 146-153.The jump in the fig. 17b (related to D = 1 m) could be because of sufficient the length of the roots which is able to improve the strength of whole surface layer thickness, but in other thicknesses, the root length was not enough to reinforcement of whole depth of layer.
The description is inserted in the result of problem.
Reviewer 2 Report
The authors presented a model called SSHV-2D, which is interesting in combing selectable theories in a single model. However, the authors should clarify some issues.
The influence of wind is not mentioned when discussing the influence of vegetation. Some factors are not easy to set up if others want to follow the same procedure. For example, the reduction factors to modified WWM. The factor seems has no obvious rule with references cited. The authors should provide a table that experiments required when trying to use SSHV-2D. Moreover, the comparison of numerical results and experiments in verification results can’t tell the experiment is done by the authors or references. When discussing the effect of root, the authors presented root density(RD). The RD is not well discussed in this study. The conclusions show that the study improves more than 90% of stability, which is not shown in the paper.Author Response
Point 1: The influence of wind is not mentioned when discussing the influence of vegetation.
Response 1: This effect was added to the model and was explained in the manuscript.
Point 2: Some factors are not easy to set up if others want to follow the same procedure. For example, the reduction factors to modified WWM. The factor seems has no obvious rule with references cited. The authors should provide a table that experiments required when trying to use SSHV-2D.
Response 2: To the determination of required parameters, field routine experiments related to each parameter should be performed. For example, the reduction factor of modified WWM may be specified by in-situ shear test.
We started a supplementary work after this paper and are performing sensitivity analysis on the parameters of the current model. The results of this analysis will determine which factors are more important. Therefore, the main tests will be specified and the process will be practical.
Point 3: Moreover, the comparison of numerical results and experiments in verification results can’t tell the experiment is done by the authors or references.
Response 3: The first sentence of section 5 was corrected. We used reported data to verification of the current model and all of the experimental data are collected by other researchers.
Point 4: When discussing the effect of root, the authors presented root density (RD). The RD is not well discussed in this study.
Response 4: The description was added.
Point 5: The conclusions show that the study improves more than 90% of stability, which is not shown in the paper.
Response 5: In example 4, this is reported that “consideration of the matric suction effect improves the SF close to two times”. This term has been extracted from fig. 15 and was stated in conclusion in other words.
Round 2
Reviewer 1 Report
the present version of the paper is satisfactory
Reviewer 2 Report
The authors have made great quality improvement of this paper. No further advice can be provide currently.